# Morpho-Molecular Characterization of Five Novel Taxa in Parabambusicolaceae (Massarineae, Pleosporales) from Yunnan, China

**DOI:** 10.3390/jof8020108

**Published:** 2022-01-24

**Authors:** Ning Xie, Rungtiwa Phookamsak, Hongbo Jiang, Yu-Jia Zeng, Haoxing Zhang, Fangfang Xu, Saisamorn Lumyong, Jianchu Xu, Sinang Hongsanan

**Affiliations:** 1Shenzhen Key Laboratory of Microbial Genetic Engineering, College of Life Sciences and Oceanography, Shenzhen University, Shenzhen 518060, China; ning.xie@szu.edu.cn; 2Honghe Center for Mountain Futures, Kunming Institute of Botany, Chinese Academy of Sciences, Honghe 654400, China; rphookamsak@outlook.com (R.P.); hongbo-j@hotmail.com (H.J.); jxu@mail.kib.ac.cn (J.X.); 3East and Central Asia Regional Office, World Agroforestry Centre (ICRAF), Kunming 650201, China; 4Centre for Mountain Futures (CMF), Kunming Institute of Botany, Kunming 650201, China; 5Center of Excellence in Fungal Research, Mae Fah Luang University, Chiang Rai 57100, Thailand; 6School of Science, Mae Fah Luang University, Chiang Rai 57100, Thailand; 7Key Laboratory of Optoelectronic Devices and Systems of Ministry of Education and Guangdong Province, College of Physics and Optoelectronic Engineering, Shenzhen University, Shenzhen 518060, China; yjzeng@szu.edu.cn (Y.-J.Z.); zhang.haoxing@szu.edu.cn (H.Z.); ff.xu@szu.edu.cn (F.X.); 8Research Center of Microbial Diversity and Sustainable Utilization, Faculty of Sciences, Chiang Mai University, Chiang Mai 50200, Thailand; scboi009@gmail.com; 9Academy of Science, The Royal Society of Thailand, Bangkok 10300, Thailand; 10Department of Biology, Faculty of Science, Chiang Mai University, Chiang Mai 50200, Thailand

**Keywords:** bambusicolous fungi, freshwater fungi, *Neomultiseptospora*, *Parabambusicola*, *Scolecohyalosporium*, taxonomy

## Abstract

Parabambusicolaceae is a well-studied family in Massarineae, Pleosporales, comprising nine genera and approximately 16 species. The family was introduced to accommodate saprobic bambusicola-like species in both freshwater and terrestrial environments that mostly occur on bamboos and grasses but are also found on different host substrates. In the present study, we surveyed and collected ascomycetes from bamboo and submerged grass across Yunnan Province, China. A biphasic approach based on morphological characteristics and multigene phylogeny demonstrated five new taxa in Parabambusicolaceae. A novel genus *Scolecohyalosporium* is introduced as a monotypic genus to accommodate *S. submersum* sp. nov., collected from dead culms of grass submerged in a freshwater stream. The genus is unique in forming filiform ascospores, which differ from other known genera in Parabambusicolaceae. Multigene phylogeny showed that the genus has a close relationship with *Multiseptospora*. Moreover, the novel monotypic genus *Neomultiseptospora*, isolated from bamboo, was introduced to accommodate *N. yunnanensis* sp. nov. *Neomultiseptospora yunnanensis* formed a separated branch basal to *Scolecohyalosporium submersum* and *Multiseptospora thailandica* with high support (100% ML, 1.00 PP). Furthermore, the newly introduced species, *Parabambusicola hongheensis* sp. nov. was also isolated from bamboo in terrestrial habitats. *Parabambusicola hongheensis* clustered with the other three described *Parabambusicola* species and has a close relationship with *P. bambusina* with significant support (88% ML, 1.00 PP). *Parabambusicola hongheensis* was reported as the fourth species in this genus. Detailed description, illustration, and updated phylogeny of Parabambusicolaceae were provided.

## 1. Introduction

Parabambusicolaceae is a family in the suborder Massarineae, order Pleosporales, class Dothideomycetes [1,2]. The family was introduced by Tanaka et al. [1] to accommodate taxa resembling the genus *Bambusicola* in Bambusicolaceae. The family is characterized by scattered or in group, immersed to erumpent, globose to subglobose, or hemispherical papillate ascomata, thin- to thick-walled peridium, composed of brown to dark brown pseudoparenchymatous cells, with heavily melanized cells towards the ostiolar neck. Asci are fissitunicate, broadly cylindrical to clavate asci, embedded in an anastomosed, narrow, branched pseudoparaphyses. Ascospores are clavate to fusiform or ellipsoidal, hyaline, frequently multi-septate [1,3].

The asexual morphs of Parabambusicolaceae are known as phoma-like coelomycetes [4] and monodictys-like or trimmatostroma-like hyphomycetes [1,5,6,7]. The coelomycetous asexual morph was reported by Phukhamsakda et al. [4] for *Neoaquastroma bauhiniae* and *N. krabiense*, which were found as saprobes on *Bauhinia variegata* L. (Fabaceae) and *Barringtonia acutangula* (Lecythidaceae) in Thailand, respectively. The coelomycetous asexual morph is characterized by pycnidial, dark brown to black, globose, uniloculate, ostiolate conidiomata, enteroblastic, phialidic, integrated conidiogenous cells, and oblong to ellipsoidal or ovoid, hyaline, aseptate, smooth-walled conidia [4]. Moreover, Tanaka et al. [1] also reported the spermatium of *Parabambusicola bambusina* from the culture of isolate KT 2637, characterized by ellipsoidal or ovoid, hyaline, aseptate, smooth-walled conidia.

The hyphomycete asexual genera in Parabambusicolaceae were referred to as *Paramonodictys*, *Paratrimmatostroma*, and *Pseudomonodictys* [5,6,7]. *Paramonodictys* and *Pseudomonodictys* are somewhat similar to *Monodictys* in having pigmented, dictyosporous conidia. However, these two genera differ from *Monodictys* due to *Paramonodictys* forming conidia from stroma [7], while the conidia of *Pseudomonodictys* form granular contents and the culture also produces red pigments on PDA, which has never been reported for *Monodictys* [5]. *Paramonodictys* is characterized by erumpent, erect, subcylindrical, or truncated-cone-form stroma, lacking conidiophores, monoblastic conidiogenous cells, and solitary, muriform, globose to subglobose, olivaceous brown to dark brown conidia [7]. *Pseudomonodictys* is characterized by semi-macronematous to macronematous, septate, branched, pale brown to brown conidiophores, holoblastic, doliiform, integrated, terminal or intercalary, indeterminate, smooth to verrucose, pale brown conidiogenous cells, and solitary, muriform, top-shaped, reddish-brown to dark brown conidia [5]. *Paratrimmatostroma* is characterized by effuse or confluent sporodochia, macronematous, or semi-macronematous, mononematous, prostrate, or erect, short, oblong to cylindrical, pale brown conidiophores, arising as lateral branches from creeping hyphae, bearing holoblastic, mono- to polyblastic, integrated, terminal conidiogenous cells and acropleurogenous, dark brown, helicoid, cylindrical, sigmoid, or reniform conidia [6].

The sexual genera in Parabambusicolaceae have similar morphological features. However, these genera can be distinguished based on multigene phylogeny. Tanaka et al. [1] accepted *Aquastroma*, *Multiseptospora*, *Parabambusicola* as well as the other two “*Monodictys*” species in Parabambusicolaceae. Li et al. [8] included the monotypic genus *Multilocularia* and Wanasinghe et al. [9] introduced *Neoaquastroma* to this family. Consequently, Phukhamsakda et al. [4] introduced two novel species in *Neoaquastroma* and included *Pseudomonodictys* in Parabambusicolaceae based on multigene phylogeny. Recently, three other monotypic genera viz. *Lonicericola*, *Paratrimmatostroma*, and *Paramonodictys* were phylogenetically affiliated in this family [6,7]. Therefore, nine genera are currently accepted in Parabambusicolaceae based on the morpho-molecular approach [3].

Members of Parabambusicolaceae have been frequently found as saprobes in both aquatic and terrestrial habitats. Species of Parabambusicolaceae occurred on a wide range of hosts, including dicots and monocots, as well as flowerless plants like ferns. However, species in this family are only restricted to Asian countries viz. China (Guizhou and Yunnan), Japan, and Thailand (Figure 1) [1,4,5,6,7,8,10,11,12,13]. Although genera in Parabambusicolaceae are well studied, most of the genera were introduced to accommodate a single species [1,5,6,7,8]. Hence, only 16 species were affiliated in this family under the assumption that more species would be introduced. In the present study, we aim to introduce two monotypic genera and a novel species in Parabambusicolaceae based on multigene phylogeny coupled with morphological distinctiveness.

## 2. Materials and Methods

### 2.1. Collection, Examination, Isolation, and Culture Preservation

Dead branches and culms of bamboo and grass were collected from Yunnan Province, China in 2021. The samples were stored in plastic Ziploc bags and returned to the laboratory for observation and examination. Fungal fruiting bodies on the host substrates were observed under an Olympus SZ61 series stereo microscope. Squash-mount slides were prepared to examine micro-morphologies and captured under a Nikon ECLIPSE Ni compound microscope connected with a Nikon DS-Ri2 camera. Indian ink was stained to check the mucilaginous sheath covered the ascospores, while congo red was stained to observe the centrum. The permanent slides contained important morphological features, were prepared by adding Lacto glycerol, and sealed using nail polish. Measurements of morphological features such as ascomata, peridium, pseudoparaphyses, asci, and ascospores were done via NIS-Elements BR 3.2 software v. 5.11.01. Photographic plates were edited and provided by Adobe Photoshop CS6 software (Adobe Systems Inc., San Jose, CA, USA).

Pure cultures were obtained from single spore isolation based on the guidelines of Senanayake et al. [14] and cultivated on potato dextrose agar (PDA) under normal light at 20–25 °C. Culture characteristics were recorded after one week and four weeks. The type specimens were deposited with permanent slides in the Herbarium of Cryptogams Kunming Institute of Botany Academia Sinica (KUN-HKAS), China, and the ex-type living culture was deposited in the Culture Collection of Kunming Institute of Botany (KUMCC). Index Fungorum numbers are provided for the newly described taxa [15].

### 2.2. DNA Extraction, Amplification, and Sequencing

The generated fungal genomic DNA was extracted from fresh fungal mycelia using Biospin Fungus Genomic DNA Extraction Kit (BioFlux^®^, Hangzhou, China) following manufacturer protocols (Hangzhou, China). The duplicated strain was also extracted from different fruiting bodies to verify the correctness of DNA sequence data using Forensic DNA Kit (Omega^®^, Norcross, GA, USA). The generated fungal genomic DNA was stored at 4 °C for PCR amplification and duplicated at −20 °C for long-term storage.

DNA amplification was conducted by polymerase chain reaction (PCR) using four phylogenetic markers viz. the internal transcribed spacers (ITS: ITS1-5.8S-ITS2), 28S large subunit rDNA (LSU), 18S small subunit rDNA (SSU), and the translation elongation factor 1-alpha (TEF1-α). Primer pairs: ITS5 and ITS4 [16], LR0R and LR5 [17], NS1 and NS4 [16], and EF1-983F and EF1-2218R [18] were used to amplify the PCR fragments of these gene regions. The final volume of the PCR reaction was 25 μL containing 2 μL of DNA template, 1 μL of each forward and reverse primers, 12.5 µL of 2× Power Taq PCR Master Mix (mixture of EasyTaqTM DNA Polymerase, dNTPs, and optimized buffer, Beijing Bio Teke Corporation (Bio Teke), Wuxi, China), and 8.5 µL of double-distilled water (ddH_2_O). The PCR thermal cycle program for ITS, LSU, SSU, and TEF1-α gene regions was set up in the following condition described by Jiang et al. [19]. PCR fragments were purified and sequenced by TsingKe Biological Technology Co., Ltd., Beijing, China. The consensus sequences of the newly generated strains are deposited in GenBank (Table 1).

### 2.3. Sequence Alignment and Phylogenetic Analyses

The consensus ITS, LSU, SSU, and TEF1-α sequences of the newly generated strains were blasted in the nucleotide BLAST search tool via the NCBI website (https://blast.ncbi.nlm.nih.gov/Blast.cgi, accessed on 25 November 2021) for searching the similar taxa. The nucleotide BLAST searches showed that the newly generated sequences had the closest similarity with species in Parabambusicolaceae. Hence, the newly generated strains were aligned with representative taxa in Parabambusicolaceae, which were retrieved from GenBank (Table 1). Individual gene alignments were aligned with MAFFT v. 7.475 [20], and the ambiguous sites were trimmed by TrimAl v. 1.3 via the web server phylemon 2 (http://phylemon.bioinfo.cipf.es/utilities.html, accessed on 25 November 2021) and improved manually where necessary using BioEdit v. 6.0.7 [21]. Improved individual gene alignments were prior analyzed by randomized axelerated maximum likelihood (RAxML) analysis via RAxML-HPC v.8 on the XSEDE (8.2.12) tool in the online web portal CIPRES Science Gateway v. 3.3 [22] using default settings but following adjustments described by Jiang et al. [19]. After checking the tree topologies of every individual gene alignment for incongruities, the combined ITS, LSU, SSU, and TEF1-α sequence dataset were then analyzed based on maximum likelihood (RAxML) and Bayesian inference (BI) analyses following commands described by Jiang et al. [19]. Phylogenetic trees were visualized on FigTree v. 1.4.0 (http://tree.bio.ed.ac.uk/software/figtree/, accessed on 25 November 2021) and edited by using Microsoft Office PowerPoint 2016 (Microsoft Inc., Redmond, WA, USA). The final sequence dataset and tree were submitted in TreeBASE (https://www.treebase.org/, accessed on 25 November 2021, submission ID: 29072).

## 3. Results

### 3.1. Phylogenetic Analyses

Primarily phylogenetic analyses based on individual genes of ITS, LSU, SSU, and TEF1-α sequence dataset from Table 1 indicated that ITS and TEF1-α are good phylogenetic markers to clarify phylogenetic resolutions of taxa in Parabambusicolaceae. In contrast, LSU and SSU have scarce phylogenetic information to evaluate the phylogenetic status of taxa, and most internal nodes lack significant support (data not shown). Thus, the combined ITS, LSU, SSU, and TEF1-α sequence dataset was conducted to clarify phylogenetic status of taxa in Parabambusicolaceae with other related families. The combined ITS, LSU, SSU, and TEF1-α sequence dataset comprises 59 strains of 46 representative species in Parabambusicolaceae and related families in Massarineae. The dataset consists of 3821 total characters, including gaps (ITS: 1–600 bp, LSU: 601–1486 bp, SSU: 1487–2889 bp, TEF1-α: 2890–3821 bp). The best scoring RAxML tree was selected to represent the phylogenetic relationships of the novel taxa with other representative taxa in Parabambusicolaceae (Figure 2), with a final ML optimization likelihood value of −23,900.440712 (ln). All free model parameters were estimated by the GTRGAMMA model, with 1354 distinct alignment patterns and 28.22% undetermined characters or gaps. Estimated base frequencies were as follows: A = 0.236175, C = 0.253070, G = 0.272964, T = 0.237791, with substitution rates AC = 1.158716, AG = 2.682148, AT = 1.418647, CG = 1.002102, CT = 6.123283, GT = 1.000000. The gamma distribution shape parameter alpha = 0.192280 and the Tree-Length = 2.440143. The final average standard deviation of split frequencies at the end of total MCMC generations was calculated as 0.008507 in BI analysis.

Overall tree topologies based on maximum likelihood (ML), and Bayesian inference (BI) analyses were similar and not significantly different. Three novel species formed well-resolved subclades within Parabambusicolaceae. *Parabambusicola hongheensis* (KUMCC 21-0410) formed a well-resolved clade with other *Parabambusicola* species and has a close relationship with *P. bambusina* (KH 139, KT 2637, H 4321) with significant support (88% ML, 1.00 PP; Figure 2). Two strains of *Neomultiseptospora yunnanensis* (KUMCC 21-0411, KUN-HKAS 122240) formed a robust clade (100% ML, 1.00 PP) basal to the novel monotypic genus *Scolecohyalosporium* and the genus *Multiseptospora* with high support (100% ML, 1.00 PP). Three representative strains of *Scolecohyalosporium submersum* sp. nov. (KUMCC 21-0412, KUMCC 21-0413, KUN-HKAS 122242) formed a robust clade (100% ML, 1.00 PP), sister to *Multiseptospora thailandica* (MFLUCC 11-0184, MFLUCC 11-0204, MFLUCC 12-0006) with significant support (77% ML, 1.00 PP).

### 3.2. Taxonomy

*Neomultiseptospora* N. Xie, Phookamsak & Hongsanan, gen. nov.

Index Fungorum number: IF 559389

Etymology: Referring to the taxon has a close phylogenetic relationship with the genus *Multiseptospora*.

Saprobic on bamboo. Sexual morph: Ascomata solitary, scattered, immersed, visible as raised, black dome-shaped on host surface, uni-loculate, hemispherical to subconical, with wedge-shaped at sides, glabrous, ostiole central or aside, with apapillate. Peridium unequally thick-walled, thick at the sides, poorly developed at the base, composed of several layers of brown to dark brown pseudoparenchymatous cells, arranged in *textura angularis* to *textura prismatica*, with palisade-like cells at sides, outer layers intermixed with host cortex. Hamathecium composed of dense, filamentous, branched, septate, cellular pseudoparaphyses, slightly constricted at the septa, anastomosed above the asci, embedded in a hyaline gelatinous matrix. Asci 8-spored, bitunicate, fissitunicate, clavate, shortly pedicellate, apically rounded, with the well-developed ocular chamber. Ascospores overlapping 1–2-seriate, hyaline, fusiform to ellipsoidal, or oblong, slightly curved, septate, smooth-walled, surrounded by a distinct, thick, mucilaginous sheath. Asexual morph: Undetermined.

Type species: *Neomultiseptospora yunnanensis* Phookamsak, Hongsanan and N. Xie

Notes: The novel genus *Neomultiseptospora* is introduced herein to accommodate the single species *N. yunnanensis* collected from dead branches of bamboo in Yunnan Province, China. Multigene phylogeny demonstrated that *Neomultiseptospora* clustered with the genera *Multiseptospora* and *Scolecohyalosporium* (Figure 2). Morphologically, *Neomultiseptospora* resembles *Multiseptospora* in having phragmosporous, hyaline, multi-septate ascospores but differs in forming raised, hemispherical to subconical ascomata, while *Multiseptospora* has globose to subglobose ascomata, covered by dark, hair-like hyphae and embedded in host tissue [10]. Additionally, *Neomultiseptospora* has a close phylogenetic relationship with *Scolecohyalosporium* but the genus is distinct from *Scolecohyalosporium* in having phragmosporous ascospores while *Scolecohyalosporium* has scolecosporous ascospores. Unfortunately, *Neomultiseptospora* is morphologically similar to the generic type of *Parabambusicola* in having hemispherical to subconical ascomata and hyaline, fusiform, 5-septate ascospores and occurring on bamboo (*Sasa* spp.) [1]. However, multigene phylogenetic analyses based on a combined ITS, LSU, SSU, and TEF1-α sequence dataset indicated the distinctiveness between these two genera.

*Neomultiseptospora yunnanensis* Phookamsak, Hongsanan and N. Xie, sp. nov.

Index Fungorum number: IF 559390, Figure 3.

Etymology: Referring to the locality, Yunnan Province, China, where the species was collected.

Holotype: KUN-HKAS 122240

Saprobic on dead branches of bamboo. Sexual morph: Ascomata 75–100 μm high, 340–450 μm diam, solitary, scattered, immersed, visible as raised, black, dome-shaped on host surface, uni-loculate, hemispherical to subconical, with wedge-shaped at sides, glabrous, ostiole central or aside, with apapillate. Peridium unequally thick-walled, 15–35 μm wide at the apex, 60–100 μm wide at sides, poorly developed at the base, composed of several layers of brown to dark brown pseudoparenchymatous cells, arranged in *textura angularis* to *textura prismatica*, with palisade-like cells at sides, outer layers intermixed with host cortex. Hamathecium composed of dense, 1–2 μm wide, filamentous, branched, septate, cellular pseudoparaphyses, slightly constricted at the septa, anastomosed above the asci, embedded in a hyaline gelatinous matrix. Asci (50–)60–90(–98) × 15–20 μm (x̅ = 76.1 × 17.5 μm, *n* = 30), 8-spored, bitunicate, fissitunicate, clavate, shortly pedicellate, apically rounded, with the well-developed ocular chamber. Ascospores (22–)23–27 × 5–8.5 μm (x̅ = 25 × 7.1 μm, *n* = 30), overlapping 1–2-seriate, hyaline, fusiform to ellipsoidal, or oblong, with rounded ends, slightly curved, (4–)5-septate, constricted at the central septum, less constricted at the other septa, smooth-walled, with small guttules, surrounded by a distinct, thick, mucilaginous sheath. Asexual morph: Undetermined.

Culture characteristics: Ascospores germinated on PDA within 24 h. Colonies on PDA reaching 27–30 mm diam after 2 weeks at room temperature (15–20 °C). Colonies dense, irregular in shape, convex to umbonate, surface smooth with an undulate edge, floccose to cottony; from above pale grey to white-grey at the margin, grey at the middle towards the center; from below white-brown at the margin, dark brown to black at the middle towards the center; not producing pigmentation on PDA.

Material examined: China, Yunnan Province, Honghe Autonomous Prefecture, Honghe County, Honghe Hani Rice Terraces (23°5′35″ N, 102°46′47″ E, 1432 + 6 msl), on the dead stem of bamboo in a terrestrial environment, 26 January 2021, R. Phookamsak, BN09A (KUN-HKAS 122240, holotype), ex-type living culture, KUMCC 21-0411.

Notes: The nucleotide BLAST search of ITS sequence showed that the closest similarity of *Neomultiseptospora yunnanensis* (KUMCC 21-0411) is “Pleosporales sp. strain 1192” (95.26% similarity, Identities = 402/422, with 1 gap) and is identical to *Multiseptospora thailandica* strain MFLUCC 11-0204 with 94.07% similarity (Identities = 428/455, with no gap), strain MFLUCC 12-0006 with 94.07% similarity (Identities = 428/455, with no gap), and strain MFLUCC 11-0183 with 94.04% similarity (Identities = 426/453, with no gap). Based on the nucleotide BLAST search of LSU sequence, *N. yunnanensis* is similar to *M. thailandica* strain MFLUCC 12-0006 (98.10% similarity, Identities = 824/840, with 3 gaps), strain MFLUCC 11-0204 (98.18% similarity, Identities = 808/823, with 2 gaps), and strain MFLUCC 11-0183 (97.91% similarity, Identities = 749/765 with 2 gaps). The closest hits based on TEF1-α sequence are *M. thailandica* strain MFLUCC 12-0006 (96.12% similarity, Identities = 817/850 with no gap), *Neoophiosphaerella sasicola* strain KT 1706 (93.16% similarity, Identities = 858/921 with no gap), and *Wettsteinina lacustris* isolate AFTOL-ID 1592 (93.12% similarity, Identities = 866/930 with no gap).

Based on a nucleotide pairwise comparison, *Neomultiseptospora yunnanensis* differs from *Multiseptospora thailandica* (MFLUCC 11-0183, type strain) in 24/416 bp of ITS (5.77%), 15/833 bp of LSU (1.8%), and 39/913 bp of TEF1-α (4.27%). Moreover, *N. yunnanensis* differs from *Scolecohyalosporium submersum* (KUMCC 21-0412) in 27/417 bp of ITS (6.47%), 16/850 bp of LSU (1.88%), and 35/923 bp of TEF1-α (4.27%). *Neomultiseptospora yunnanensis* can be distinguished from *M. thailandica* in having fusiform to ellipsoidal, or oblong, (4–)5-septate ascospores, whereas *M. thailandica* has typical fusiform to vermiform, 10–11-septate ascospores [10]. *Neomultiseptospora yunnanenis* clearly distinct from *S. submersum* due to *S. submersum* having filiform, multi-septate (up to 20 septa) ascospores.

*Parabambusicola hongheensis* H.B. Jiang and Phookamsak, sp. nov.

Index Fungorum number: IF 559391, Figure 4.

Etymology: The specific epithet “*hongheensis*” refers to the location, Honghe, Yunnan Province of China, where the new species was collected.

Holotype: KUN-HKAS 122607

Saprobic on dead bamboo culms in a terrestrial environment. Sexual morph: Ascomata 200–300 μm high, 340–500 μm diam, gregarious, immersed under host epidermis, raised, dark brown to black, coriaceous, subglobose in surface view, hemispherical or conical with a flattened base in cross-section, uni-loculate, ostiolate, with a pore-like opening at the top. Peridium 20–75 μm wide, comprising dark brown to pale brown cells arranged in *textura angularis*, intermixed with host tissue. Hamathecium composed of 1.3–2.5 μm wide, filamentous, septate, anastomosed pseudoparaphyses, embedded in a hyaline gelatinous matrix. Asci 115–140 × 30–35 μm (x̅ = 128 × 32.5 μm, *n* = 20), 8-spored, bitunicate, fissitunicate, broadly cylindrical to clavate, with a swollen end, apically rounded, with an inconspicuous ocular chamber. Ascospores 50–55 × 12–15 μm (x̅ = 53 × 13.5 μm, *n* = 20), overlapping 2–3-seriate, subhyaline to hyaline, fusiform to vermiform, narrower towards the lower part, enlarged at the 2nd cell from the apex, slightly curved, 3–4-septate, slightly constricted at the septa, smooth-walled, with small, multi-guttulate, surrounded by a thin entire sheath. Asexual morph: Undetermined.

Culture characteristics: Ascospores germinating on PDA within 24 h and germ tubes produced from both cells of ascospores. Colonies on PDA reaching 30 mm diam after 3 weeks at room temperature (10–20 °C). Colonies dense, circular, flattened, floccose; from above pale brown at the middle, pale white at the margin; from below dark brown to black; not producing pigmentation on agar medium. Mycelium superficial to immersed in media, with branched pale grey to black, septate, smooth hyphae.

Material examined: China, Yunnan Province, Honghe Autonomous Prefecture, Honghe County, Baohua Village (23°15′19″ N, 102°21′31.56″ E, altitude 1816.52 msl), on dead culms of bamboo in a terrestrial environment, 28 October 2020, H.B. Jiang, HONGHE015 (KUN-HKAS 122607, holotype), ex-type living culture, KUMCC 21-0410.

Notes: The NCBI nucleotide BLAST search of ITS sequence indicated that *Parabambusicola hongheensis* (KUMCC 21-0410) is similar to *P. bambusina* strain KT 2637 with 98.30% similarity (Identities = 521/530, with 3 gaps), and *P. bambusina* strain H4321 with 98.11% similarity (Identities = 520/530, with 4 gaps). Nucleotide BLAST results of LSU sequence showed that *P. hongheensis* is similar to “*Parabambusicola* sp. strain GZCC19-0505” (100% similarity, Identities = 783/783, with no gap), *P. bambusina* strain H4321 (99.64% similarity, Identities = 840/843, with 1 gap), and *P. thysanolaenae* strain KUMCC 18-0147 (99.53% similarity, Identities = 839/843, with 1 gap).

A nucleotide pairwise comparison of ITS and LSU sequences indicated that *Parabambusicola hongheensis* differs from *P. bambusina* (H 4321, KT 2637, KH 139) in 9/530 bp (1.7%), and 2/843 bp (0.2%), respectively. *Parabambusicola hongheensis* differs from *P. aquatica* (MLFUCC 18-1140, type strain) in 40/444 bp of ITS (9%), and 9/836 bp of LSU (1.1%). The species also differs from *P. thysanolaenae* (KUMCC 18-0147, type strain) in 44/446 bp of ITS (9.9%), and 4/843 bp of LSU (0.4%). *Parabambusicola hongheensis* typically morphological resembles species in *Parabambusicola* in forming gregarious, immersed to erumpent, raised, hemispherical ascomata, with flattened base, broadly cylindrical to clavate asci, with subsessile to short pedicel and phragmosporous, hyaline, fusiform ascospores [1,6,12]. *Parabambusicola hongheensis* can be distinguished from the other three *Parabambusicola* species in having 3–4-septate ascospores, while the other *Parabambusicola* species have 5–(6–7)-septate ascospores [1,6,12].

*Scolecohyalosporium* N. Xie, Phookamsak and Hongsanan, gen. nov.

Index Fungorum number: IF 559392

Etymology: Referring to the taxon forming hyaline scolecosporous ascospores on grass (Poaceae).

Saprobic on stems of unidentified grass submerged in a small stream. Sexual morph: Ascomata solitary, scattered. sometimes clustered, erumpent through host cortex, becoming superficial, uni-loculate, conical to ovoid, glabrous, rough-walled, ostiole central, with truncate apex. Peridium thick-walled, of unequal thickness, thick at sides towards the apex, thin at the base, composed of several layers of small, dark brown to black, pseudoparenchymatous cells, arranged in a *textura angularis*, with carbonaceous cells at the ostiole, outer layers intermixed with the host tissues. Hamathecium composed of dense, filamentous, branched, septate, cellular pseudoparaphyses, constricted at the septa, anastomosed among the asci, embedded in a gelatinous matrix. Asci 8-spored, bitunicate, fissitunicate, long cylindrical, shortly pedicellate, apically rounded with the indistinct ocular chamber. Ascospores spirally arranged within the ascus, hyaline, filiform, narrower towards the end cells, multi-septate, not constricted at the septa, smooth-walled. Asexual morph: Undetermined.

Type species: *Scolecohyalosporium submersum* Phookamsak, Hongsanan and N. Xie

Notes: *Scolecohyalosporium* is introduced herein as a monotypic genus to accommodate *S. submersum* sp. nov. The species was isolated from dead stems of grass submerged in a freshwater stream in Yunnan, China. *Scolecohyalosporium* has a unique character that can be distinguished from other sexual genera in Parabambusicolaceae in forming scolecosporous ascospores. *Scolecohyalosporium* morphological resembles genera *Neoophiosphaerella* and *Poaceascoma* in Lentitheciaceae. However, *Neoophiosphaerella* can be distinguished from *Scolecohyalosporium* in having subglobose to hemispherical ascomata, with thin-walled, brown cells of peridium, and broader cylindrical asci [1], while the type of *Poaceascoma* formed setose ascomata [23].

*Scolecohyalosporium submersum* Phookamsak, Hongsanan and N. Xie, sp. nov.

Index Fungorum number: IF 559393, Figure 5.

Etymology: Referring to the habitat of the taxon, submerged in a small freshwater stream.

Holotype: KUN-HKAS 122242

Saprobic on stems of unidentified grass submerged in a small stream. Sexual morph: Ascomata 370–480 μm high, 380–600 μm diam, solitary, scattered. sometimes clustered, erumpent through host cortex, becoming superficial, uni-loculate, conical to ovoid, glabrous, rough-walled, ostiole central, with truncate apex. Peridium 50–90 μm wide at sides, 20–40 μm wide at the base, thick-walled of unequal thickness, thick at sides towards the apex, thin at the base, composed of several layers of small, dark brown to black, pseudoparenchymatous cells, arranged in a *textura angularis*, with carbonaceous cells at the ostiole, outer layers intermixed with the host tissues. Hamathecium composed of dense, 1.8–3.6 μm wide, filamentous, tapering towards the apex, branched, septate, cellular pseudoparaphyses, constricted at the septa, anastomosed among the asci, embedded in a gelatinous matrix. Asci (200–)250–300(–370) × 7–9(–11) μm (x̅ = 274.6 × 8.2 μm, *n* = 30), 8-spored, bitunicate, fissitunicate, long cylindrical, shortly pedicellate, apically rounded with the indistinct ocular chamber. Ascospores (230–)260–285(–315) × 1.5–2.2 μm (x̅ = 267.5 × 1.9 μm, *n* = 30), spirally arranged within the ascus, hyaline, filiform, narrower towards the end cells, multi-septate, up to 20 septa, not constricted at the septa, smooth-walled. Asexual morph: Undetermined.

Culture characteristics: Ascospores germinated on PDA within 24 h. Colonies on PDA reaching 18 mm diam after 2 weeks at room temperature (15–20 °C). Colonies dense, circular, low convex, surface smooth with an entire edge, floccose to cottony; from above pale grey at the margin, grey to dark grey at the center; from below white at the margin, black at the middle, pale grey at the center; not producing pigmentation on PDA.

Material examined: China, Yunnan Province, Xishuangbanna Dai Autonomous Prefecture, Mengla County, Bubeng (21°36′30.13″ N, 101°35′52.54″ E, 664 ± 5 msl), on stems of unidentified grass submerged in a freshwater stream, 27 April 2021, R. Phookamsak, BB21-006 (KUN-HKAS 122242, holotype), ex-type living culture, KUMCC 21-0412, KUMCC 21-0413.

Notes: The NCBI nucleotide BLAST search of ITS sequence indicated that *Scolecohyalosporium submersum* (KUMCC 21-0412) is similar to “Pleosporales sp. strain 1192” with 96.43% similarity (Identities = 432/448, with 2 gaps), *Parabambusicola aquatica* strain MFLUCC 18-1140 with 95.58% similarity (Identities = 238/249, with 2 gaps), and *Neoaquastroma krabiense* strains MFLUCC 16-0419 with 93.65% similarity (Identities = 236/252, with 2 gaps). The nucleotide BLAST search of LSU sequence indicated that the closest hit of *S. submersum* (KUMCC 21-0412) is *Multiseptospora thailandica* strain MFLUCC 11-0204 (98.80% similarity, Identities = 823/833, with 1 gap), strain MFLUCC 12-0006 (98.71% similarity, Identities = 839/850, with 2 gaps), and strain MFLUCC 11-0183 (98.56% similarity, Identities = 754/765 with 1 gap). The most similar taxa based on NCBI BLAST search of TEF1-α sequence are *M. thailandica* strain MFLUCC 12-0006 (96.59% similarity, Identities = 821/850 with no gap), *Wettsteinina lacustris* isolate AFTOL-ID 1592 (93.44% similarity, Identities = 869/930 with no gap), and “*Monodictys* sp. strain JO 10 EF45a” (93.20% similarity, Identities = 863/926 with 1 gap). According to Tanaka et al. [1], *Wettsteinina lacustris* isolate AFTOL-ID 1592 was treated in Lentitheciaceae. Therefore, the isolate was not included in the present phylogenetic analyses. Based on a nucleotide pairwise comparison of ITS, LSU, and TEF1-α sequences, *S. submersum* differs from *M. thailandica* (MFLUCC 11-0183, type strain) in 25/419 bp (6%), 11/851 bp (1.3%), 32/895 bp (3.6%), respectively.

## 4. Discussion

Parabambusicolaceae is a well-studied family based upon the morphological characterization and molecular phylogeny. Even though the family was introduced in 2015 by K. Tanaka and his colleagues to initially accommodate two novel monotypic genera in Massarineae, the number of genera and species in Parabambusicolaceae has continually increased in recent years [4,6,7,8,9,11,12,13]. In this study, we establish the other two novel genera, *Neomultiseptospora* and *Scolecohyalosporium* as well as the novel species *Parabambusicola hongheensis* in Parabambusicolaceae based on multigene phylogeny coupled with morphological differences. Hence, there are 11 genera and 19 phylogenetic species in this family.

Most genera in Parabambusicolaceae were introduced as monotypic genera, and many genera are currently accommodated as putative species, including *Aquastroma*, *Multilocularia*, *Neomultiseptospora*, *Paramonodictys*, *Paratrimmatostroma*, *Pseudomonodictys*, and *Scolecohyalosporium*. *Aquastroma*, *Multilocularia*, *Neomultiseptospora*, and *Scolecohyalosporium* are only represented by the sexual morph, while *Paramonodictys*, *Paratrimmatostroma*, and *Pseudomonodictys* are represented by the hyphomycetous asexual morph. Due to these genera being represented by a single species, the phylogenetic relationships with the related genera are sometimes not well-resolved. This may be because of a few genetic information and taxon sampling.

Multigene phylogenetic analyses demonstrated that *Lonicericola*, *Multilocularia*, *Neoaquastroma*, *Neomultiseptospora*, *Parabambusicola*, *Paramonodictys*, *Paratrimmatostroma*, and *Scolecohyalosporium* form well-resolved subclade (up to 70% ML and 0.95 PP support; Figure 2) in this study. However, the phylogenetic relationships among *Lonicericola*, *Paratrimmatostroma*, and *Pseudomonodictys* are not well-resolved. These three genera clade together with low supported values in this study that concurred with Hyde et al. [7]. *Paratrimmatostroma* (trimmatostroma-like), and *Pseudomonodictys* (monodictys-like) are represented by their hyphomycetous asexual morph, whereas *Lonicericola* is represented by massarina-like sexual morph. Hence, the morphological features of *Lonicericola* cannot be compared with *Paratrimmatostroma*, and *Pseudomonodictys*, while *Paratrimmatostroma*, and *Pseudomonodictys* have morphological differences. Moreover, *Aquastroma* (sexual morph) also clusters with *Paramonodictys* (asexual morph), and “*Monodictys* sp. JO 10” with low support. Therefore, more described holomorph species are necessary in these genera to support their generic status.

*Multiseptospora* was introduced as a monotypic genus by Liu et al. [10] to accommodate *M. thailandica*, isolated from the dead stem of grass (*Thysanolaena maxima*) in Thailand. Subsequently, Li et al. [8] introduced the second species, *M. thysanolaenae*, to the genus. *Multiseptospora thysanolaenae* morphological resembles the type species *M. thailandica* but differs in forming brown ascospores [8]. Phylogenetic relationships between *M. thysanolaenae* and *M. thailandica* were not well-resolved in Li et al. [8] and the species always formed a separated branch basal to *Parabambusicola* in subsequent studies [4,6,7,9,11]. Due to most sexual genera in Parabambusicolaceae having similar morphology, *M. thysanolaenae* may not be congeneric with *M. thailandica* based on phylogenetic evidence. However, the genetic information of *M. thysanolaenae* is not well-verified (R. Phookamsak, pers. comm.). The ex-type strain of this species needs to be re-sequenced for further evaluating its generic status.

Most sexual genera in Parabambusicolaceae have similar morphology of the ascospores. However, these genera can be distinguished from each other based on the other morphological features such as the shape of ascomata, peridial structure, host occurrences, and habitats, and the affinities of these genera were also supported by phylogenetic evidence. We thus provide the keys to genera in Parabambusicolaceae for better clarifying the generic resolution.

Key to genera of Parabambusicolaceae

1. Asexual morph ................................................................................................................ 2

1. Sexual morph ................................................................................................................... 5

2. Hyphomycetous asexual morph ................................................................................... 3

2. Coelomycetous asexual morph, the sexual morph having obovoid to clavate asci, occurred on dicots .................................................................................. ***Neoaquastroma***

3. Monodictys-like ............................................................................................................... 4

3. Trimmatostroma-like, with varied in shape, brown conidia, occurred on a ferns ........................................................................................................ ***Paratrimmatostroma***

4. Conidia globose or subglobose, with multi cells, arising from subcylindrical or truncated-cone-form stroma, lacking conidiophores ............................... ***Paramonodictys***

4. Conidia acropleurogenous, subglobose to ellipsoidal, with 1–2 cells, irregular in shapes with several cells, arising from semi-macronematous to macronematous conidiophores ...................................................................................... ***Pseudomonodictys***

5. Ascospores phragmosporous, fusiform to ellipsoidal or vermiform ..................... 6

5. Ascospores scolecosporous, filiform ........................................ ***Scolecohyalosporium***

6. Ascomata solitary to gregarious, uni-loculate ............................................................ 7

6. Ascomata immersed in the stroma, multi-loculate, clavate asci with short to long pedicel and hyaline, ellipsoidal, (1–)3(–4)-septate ascospores ........... ***Multilocularia***

7. Ascomata globose to subglobose .................................................................................. 8

7. Ascomata subconical to hemispherical ...................................................................... 10

8. Asci broadly cylindrical to cylindric-clavate, with subsessile to short pedicel ...... 9

8. Asci clavate, with short pedicel, clavate to fusiform, 6–8-septate ascospores, occurred on submerged twigs of dicot .......................................................... ***Aquastroma***

9. Ascomata glabrous, with minute papillate, (8–)9-septate ascospores, occurred on dicots .............................................................................................................. ***Lonicericola***

9. Ascomata setose, covered by tufts of hyphae, apapillate, 10–11-septate ascospores, occurred on monocots ......................................................................... ***Multiseptospora***

10. Ascomata hemispherical to subconical, poorly-developed at the base, fusiform to ellipsoidal, or oblong ascospores, with rounded end cells, slightly constricted at the central septum, less constricted at the other septa .................. ***Neomultiseptospora***

10. Ascomata subglobose to hemispherical or subconical, fusiform to vermiform ascospores with acute ends, frequently not constricted at the septa, covered by a thin entire sheath ....................................................................................... ***Parabambusiccola***

## Figures and Tables

**Figure 1 jof-08-00108-f001:**
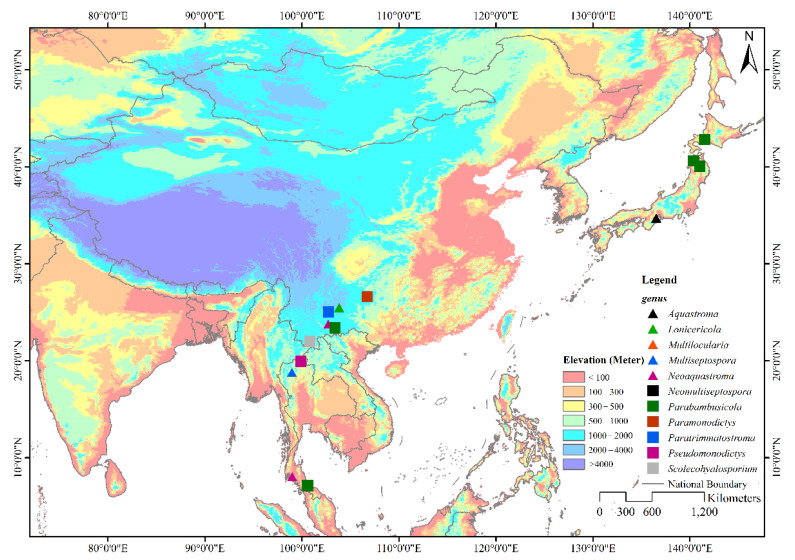
Geographical distribution of genera in Parabambusicolaceae. The eleven genera in Parabambusicolaceae that distribute in Asian countries are shown with different symbols on the map.

**Figure 2 jof-08-00108-f002:**
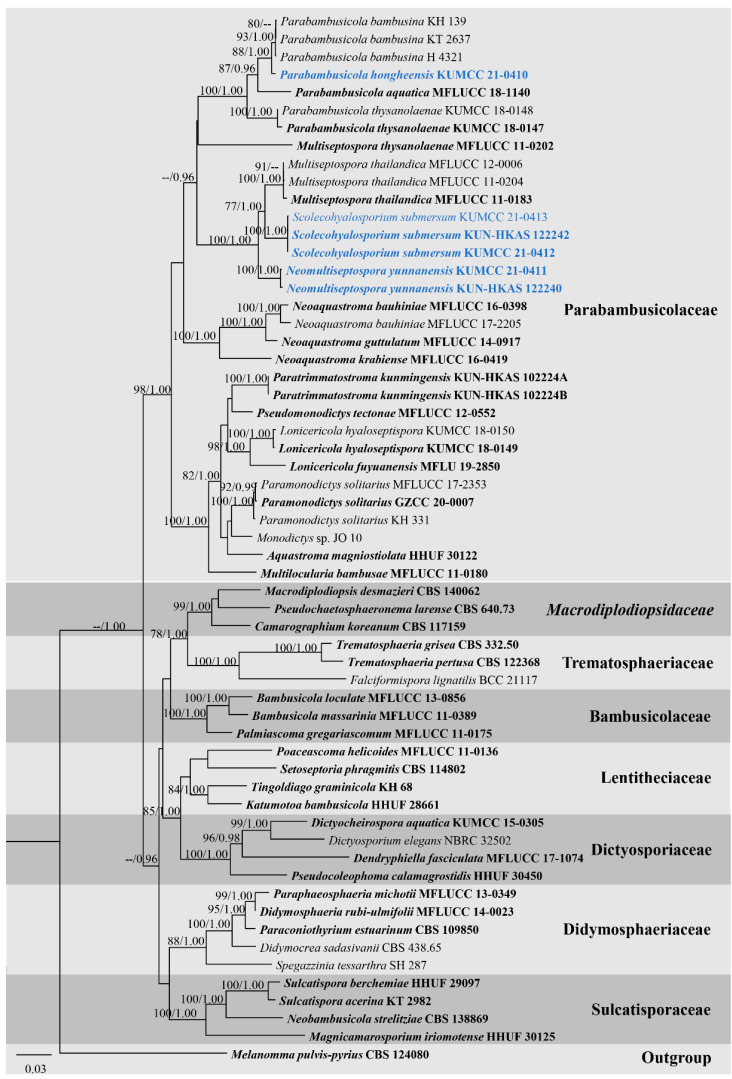
RAxML tree based on ITS, LSU, SSU, and TEF1-α sequence matrix representing the phylogenetic relationships of taxa in Parabambusicolaceae and related families in Massarineae. The tree is rooted in *Melanomma pulvis-pyrius* (CBS 124080). Bootstrap support values for ML equal to or greater than 70% and the Bayesian posterior probabilities equal to or higher than 0.95 PP are indicated above the nodes as ML/PP. Ex-type strains are in bold and the new taxa are indicated in blue.

**Figure 3 jof-08-00108-f003:**
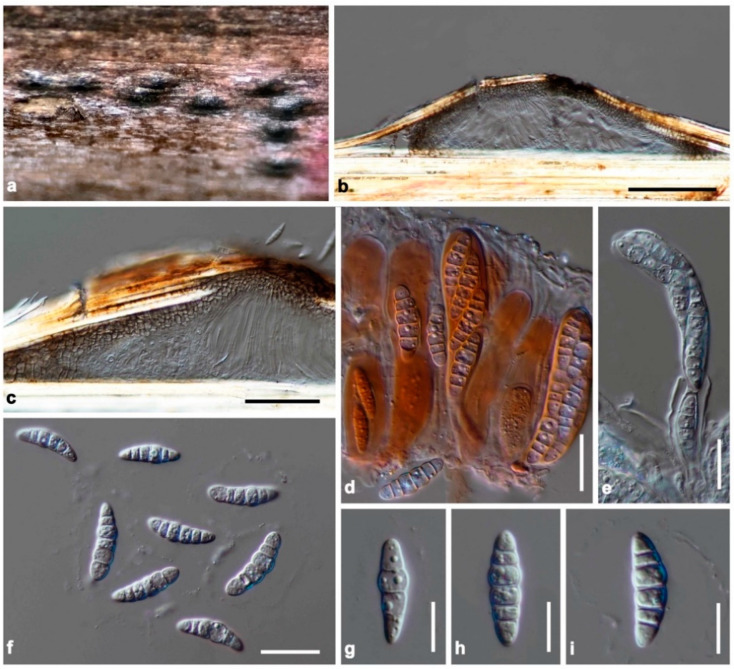
*Neomultiseptospora yunnanensis* (KUN-HKAS 122240, holotype). (**a**) The appearance of ascomata on host substrate; (**b**) vertical section of ascoma; (**c**) peridium; (**d**) asci embedded in cellular pseudoparaphyses stained with congo red; (**e**) fissitunicate ascus; (**f**–**i**) ascospores. Scale bars: (**b**) = 100 μm, (**c**) = 50 μm, (**d**–**f**) = 20 μm, (**g**–**i**) = 10 μm.

**Figure 4 jof-08-00108-f004:**
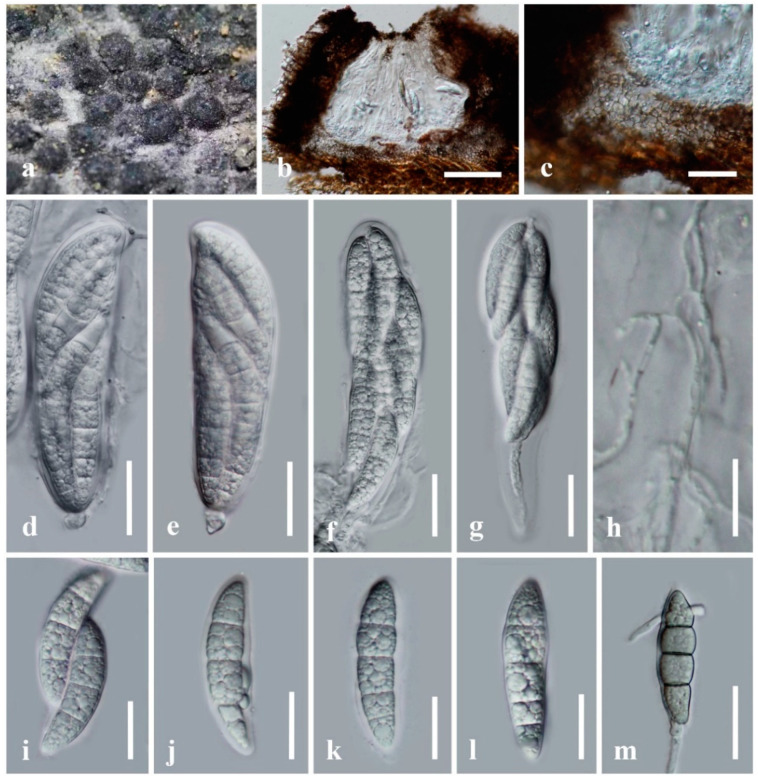
*Parabambusicola hongheensis* (KUN-HKAS 122607, holotype). (**a**) The appearance of ascomata on host substrate; (**b**) vertical section of ascoma; (**c**) peridium; (**d**–**g**) asci; (**h**) pseudoparaphyses; (**i**–**l**) ascospores; (**m**) germinated ascospore. Scale bars: (**b**) = 100 μm, (**c**–**g**,**m**) = 30 μm, (**i**–**l**) = 20 μm, (**h**) = 15 μm.

**Figure 5 jof-08-00108-f005:**
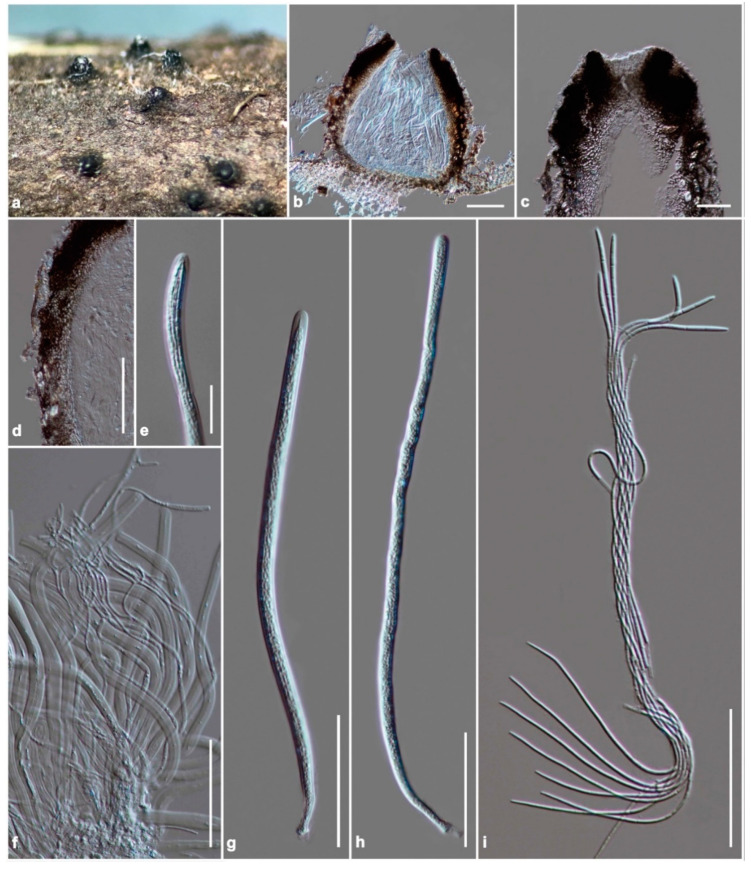
*Scolecohyalosporium submersum* (KUN-HKAS 122242, holotype). (**a**) The appearance of ascomata on host substrate; (**b**) vertical section of ascoma; (**c**) section through ostiole; (**d**) peridium; (**e**) fissitunicate at the apex of the ascus; (**f**) pseudoparaphyses; (**g**,**h**) asci; (**i**) ascospores. Scale bars: (**b**,**d**) = 100 μm, (**c**,**f**–**i**) = 50 μm, (**e**) = 20 μm.

**Table 1 jof-08-00108-t001:** Taxa used in this study and their GenBank accession numbers. The ex-type and ex-epitype strains are indicated by superscript “T”. The newly generated sequences are indicated in bold.

*Species Name*	*Strain No.*	*GenBank Accession Numbers*			
		ITS	LSU	SSU	TEF1-α
** *Aquastroma magniostiolata* ^T^ **	HHUF 30122	NR_153583	NG_056936	NG_061000	AB808486
*Bambusicola loculata* ^T^	MFLUCC 13-0856	NR_153609	NG_069267	NG_065061	KP761724
*Bambusicola massarinia* ^T^	MFLUCC 11-0389	NR_121548	NG_058658	NG_061198	KP761725
*Camarographium koreanum* ^T^	CBS 117159	JQ044432	JQ044451	n/a	n/a
*Dendryphiella fasciculata* ^T^	MFLUCC 17-1074	NR_154044	NG_059177	n/a	n/a
*Dictyocheirospora aquatica* ^T^	KUMCC 15-0305	NR_154030	KY320513	n/a	n/a
*Dictyosporium elegans*	NBRC 32502	DQ018087	DQ018100	DQ018079	n/a
*Didymocrea sadasivanii*	CBS 438.65	MH858658	DQ384103	DQ384074	n/a
*Didymosphaeria rubi-ulmifolii* ^T^	MFLUCC 14-0023	n/a	KJ436586	NG_063557	n/a
*Falciformispora lignatilis*	BCC 21117	KF432942	GU371826	GU371834	GU371819
*Katumotoa bambusicola* ^T^	HHUF 28661	NR_154103	NG_059386	NG_060989	AB539108
*Lonicericola fuyuanensis* ^T^	MFLU 19-2850	NR_172419	NG_073809	NG_070329	MN938324
*Lonicericola hyaloseptispora* ^T^	KUMCC 18-0149	NR_164294	NG_066434	NG_067680	n/a
*Lonicericola hyaloseptispora* ^T^	KUMCC 18-0150	MK098194	MK098200	MK098206	MK098210
*Macrodiplodiopsis desmazieri* ^T^	CBS 140062	NR_132924	NG_058182	n/a	n/a
*Magnicamarosporium iriomotense* ^T^	HHUF 30125	NR_153445	NG_059389	NG_060999	AB808485
*Melanomma pulvis-pyrius* ^T^	CBS 124080	MH863349	MH874873	GU456302	GU456265
*Monodictys* sp.	JO 10	n/a	AB807552	AB797262	AB808528
*Multilocularia bambusae* ^T^	MFLUCC 11-0180	NR_148099	NG_059654	NG_061229	KU705656
*Multiseptospora thailandica* ^T^	MFLUCC 11-0183	NR_148080	NG_059554	KP753955	KU705657
*Multiseptospora thailandica*	MFLUCC 11-0204	KU693447	KU693440	KU693444	KU705659
*Multiseptospora thailandica*	MFLUCC 12-0006	KU693448	KU693441	KU693445	KU705660
*Multiseptospora thysanolaenae* ^T^	MFLUCC 11-0202	n/a	NG_059655	NG_063600	KU705658
*Neoaquastroma bauhiniae* ^T^	MFLUCC 16-0398	NR_165217	NG_067814	NG_070696	MH028247
*Neoaquastroma bauhiniae*	MFLUCC 17-2205	MH025953	MH023320	MH023316	MH028248
*Neoaquastroma guttulatum* ^T^	MFLUCC 14-0917	KX949739	KX949740	KX949741	KX949742
*Neoaquastroma krabiense* ^T^	MFLUCC 16-0419	NR_165218	NG_067815	NG_067670	MH028249
*Neobambusicola strelitziae* ^T^	CBS 138869	NR_137945	NG_058125	n/a	MG976037
** *Neomultiseptospora yunnanensis* ^T^ **	**KUMCC 21-0411**	**OL898884**	**OL898925**	**OL898890**	**OL964282**
** *Neomultiseptospora yunnanensis* ^T^ **	**KUN-HKAS 122240**	**OL898885**	**OL898926**	**OL898891**	**OL964283**
*Palmiascoma gregariascomum* ^T^	MFLUCC 11-0175	NR_154316	NG_059557	KP753958	n/a
*Parabambusicola aquatica* ^T^	MFLUCC 18-1140	NR_171877	NG_073791	n/a	n/a
*Parabambusicola bambusina*	H 4321	n/a	AB807536	AB797246	AB808511
*Parabambusicola bambusina*	KH 139	n/a	AB807537	AB797247	AB808512
*Parabambusicola bambusina*	KT 2637	n/a	AB807538	AB797248	AB808513
** *Parabambusicola honghensis* ^T^ **	**KUMCC 21-0410**	**OL898880**	**OL898921**	**OL898886**	**n/a**
*Parabambusicola thysanolaenae* ^T^	KUMCC 18-0147	NR_164044	NG_066435	NG_067681	MK098209
*Parabambusicola thysanolaenae* ^T^	KUMCC 18-0148	MK098193	MK098198	MK098202	MK098211
*Paraconiothyrium estuarinum* ^T^	CBS 109850	NR_166007	MH874432	AY642522	n/a
*Paraphaeosphaeria michotii* ^T^	MFLUCC 13-0349	NR_155640	NG_059522	KJ939285	n/a
*Paramonodictys solitarius* ^T^	GZCC 20-0007	MN901152	MN897835	MN901118	MT023012
*Paramonodictys solitarius*	MFLUCC 17-2353	MT627707	MN913703	MT864299	MT954397
*Paramonodictys solitarius*	KH 331	n/a	AB807553	AB797263	AB808529
*Paratrimmatostroma kunmingensis* ^T^	KUN-HKAS 102224A	MK098192	MK098196	MK098204	MK098208
*Paratrimmatostroma kunmingensis* ^T^	KUN-HKAS 102224B	MK098195	MK098201	MK098207	n/a
*Poaceascoma helicoides* ^T^	MFLUCC 11-0136	NR_154317	NG_059565	NG_061205	KP998461
*Pseudochaetosphaeronema larense* ^T^	CBS 640.73	NR_132038	NG_057978	NG_061147	KF015684
*Pseudocoleophoma calamagrostidis* ^T^	HHUF 30450	NR_154375	NG_059804	NG_061264	LC014614
*Pseudomonodictys tectonae* ^T^	MFLUCC 12-0552	n/a	NG_059590	NG_061213	KT285571
** *Scolecohyalosporium submersum* ^T^ **	**KUMCC 21-0412**	**OL898883**	**OL898924**	**OL898889**	**OL964281**
** *Scolecohyalosporium submersum* ^T^ **	**KUMCC 21-0413**	**OL898881**	**OL898922**	**OL898887**	**OL964279**
** *Scolecohyalosporium submersum* ^T^ **	**KUN-HKAS 122242**	**OL898882**	**OL898923**	**OL898888**	**OL964280**
*Setoseptoria phragmitis* ^T^	CBS 114802	KF251249	KF251752	n/a	KF253199
*Spegazzinia tessarthra*	SH 287	n/a	AB807584	AB797294	AB808560
*Sulcatispora acerina* ^T^	KT 2982	LC014597	LC014610	LC014605	LC014615
*Sulcatispora berchemiae* ^T^	HHUF 29097	NR_153444	NG_059390	NG_064843	AB808509
*Tingoldiago graminicola* ^T^	KH 68	n/a	AB521743	AB521726	AB808561
*Trematosphaeria grisea* ^T^	CBS 332.50	NR_132039	NG_057979	NG_062930	KF015698
*Trematosphaeria pertusa* ^T^	CBS 122368	NR_132040	NG_057809	n/a	n/a

## Data Availability

All data availability was mentioned in the manuscript. The novel taxa were registered in Index Fungorum (http://www.indexfungorum.org/Names/Names.asp, accessed on 18 December 2021) including Index Fungorum numbers IF 559389, IF 559390, IF 559391, IF 559392 and IF 559393. Final alignment and phylogenetic tree were deposited in TreeBase (https://www.treebase.org/ (accessed on 25 November 2021) with submission ID: 29072) and the newly generated sequences were deposited in GenBank (https://www.ncbi.nlm.nih.gov/genbank/submit/, accessed on 25 November 2021) followed as ITS: OL898884, OL898885, OL898880, OL898883, OL898881, OL898882; LSU: OL898925, OL898926, OL898921, OL898924, OL898922, OL898923; SSU: OL898890, OL898891, OL898886, OL898889, OL898887, OL898888; TEF1-α: OL964282, OL964283, OL964281, OL964279, OL964280.

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
