# Peer review of "Morpho-Molecular Characterization of Five Novel Taxa in Parabambusicolaceae (Massarineae, Pleosporales) from Yunnan, China"

_jof, 2022, doi:10.3390/jof8020108_

Round 1
Reviewer 1 Report
The manuscript deals with an interesting subject, the re- classification of fungal families and introduced some nova species and genera.
This paper is suitable for publication in Jof journal after minor corrections in this separate sheet
Abstract
1- Line (38)… is --------- was
2- Line (43).... is---------- was
3- Line (45)… was ------were
Introduction
1- Some sentences are so long, could be reduced.
2- The aim of the present work must be included at the end of the introduction.
3- The map is not cleared.
Results
1- Table (1) must be involved in chapter results
2- Figure (2) the map is so crowded
3- All species and genera must be written in italic name.
Conclusion
I recommend acceptance of this article for publication in the Journal of Fungi
Author Response
Dear Reviewer
Thank you very much for your comments and support to improve this work.
I did revise all the point you mentioned in the previous version.
Please see the attachment for our respond to your suggestions.
:-)
Best Regards
Hongsanan

Reviewer 2 Report
The submitted manuscript by Xie et al investigated several ascomycetes from bamboo and submerged grass collected in Yunnan province, China, and reported two novel genus Scolecohyalosporium and Neomultiseptospora (isolated from submerged grass and bamboo respectively) which were both belong to Parabambusicolaceae family and a new Parabambusicola species which is close to Parabambusicola bambusina. Using morphological and molecular methods, the authors provided detailed morphological traits and figures of newly reported species, and conducted phylogenetic analyses based on ITS, LSU, SSU and TEF1-α gene markers. The work was well done. I believe this manuscript is of great interesting and substantially contribute to enrich the knowledge of Parabambusicolaceae family, thus I would be willing to recommend it for the publication in Journal of Fungi.
Author Response
Dear Reviewer,
We appreciate the time and effort that the you have dedicated to providing valuable feedback on our manuscript.
Best Regards
Hongsanan
Reviewer 3 Report
As i read the work is Good written and big efforts and interested, Fungi is big world need to be discovered and explored
My opinion this work is accepted
Author Response
Dear Reviewer,
We appreciate the time and effort that you have dedicated to providing valuable feedback on our manuscript.
Best Regards
Hongsanan